# Dietary Modelling to Explore the Impact of Potassium Chloride Replacement for Sodium in Bread for Adults with Chronic Kidney Disease

**DOI:** 10.3390/nu13072472

**Published:** 2021-07-20

**Authors:** Rebecca Morrison, Jordan Stanford, Kelly Lambert

**Affiliations:** 1School of Medicine, Faculty of Science Medicine and Health, University of Wollongong, Wollongong, NSW 2522, Australia; rm480@uowmail.edu.au (R.M.); js096@uowmail.edu.au (J.S.); 2Illawarra Health and Medical Research Institute, Wollongong, NSW 2522, Australia

**Keywords:** dietary modelling, chronic kidney disease, salt, food supply, public health, potassium, bread, food labelling

## Abstract

Food manufacturers are increasingly substituting potassium chloride (KCl) in food products so as to reduce the sodium chloride content. Bread and bread products are common staple foods in many Western households and are a target for recipe reformulation using KCl. Given that chronic kidney disease (CKD) is a medical condition of global importance that requires dietary potassium restriction in the later stages, we sought to evaluate the impact and safety of varying levels of KCl substitution in bread products. We undertook a secondary analysis of dietary data from the National Nutrition and Physical Activity Survey 2011–2012 for 12,152 participants (154 participants with CKD). The sodium chloride content in bread and bread-based products was substituted with 20%, 30%, and 40% of KCl. The contribution of these alterations in the dietary potassium intake to the total daily potassium intake were then examined. The replacement of sodium in bread with varying amounts of KCl (20%, 30%, and 40%) resulted in one third of people with CKD exceeding the safe limits for dietary potassium consumption (31.8%, 32.6%, and 33%, respectively). KCl substitution in staple foods such as bread and bread products have serious and potentially fatal consequences for people who need to restrict dietary potassium. Improved food labelling is required for consumers to avoid excessive consumption.

## 1. Introduction

Sodium is the primary cation involved in fluid balance [1]. It plays key roles in fundamental physiological activities such as controlling extracellular fluid movement and manipulating the cellular membrane potential [1]. Most dietary sodium consumed by individuals is in the form sodium chloride (NaCl), because of the stability of the ion pairing, and it is therefore a common presentation in food recipes [2]. Excessive intake of dietary sodium leads to hypertension and cardiovascular disease, which is a leading cause of chronic kidney disease (CKD) globally.

The positive association between high sodium consumption and an increased risk of cardiovascular disease (CVD) has been well established over the past few decades [3]. To combat this, the World Health Organisation has strongly recommended individuals limit their dietary sodium intake to less than 2 g/day of sodium (or 5 g/day of salt) for adults [4]. However, the current global average intake is almost double this amount at approximately 3.95 g of sodium per day [5]. This excessive consumption of sodium is expected to contribute to the rise in the prevalence of hypertension to approximately 1.5 billion adults by the year 2025 [3].

Excessive sodium intake is also a major problem in Australia. The approximate sodium intake of Australians has been estimated to be approximately 3.74 g of sodium per day (9.6 g of salt), which is almost double the WHO recommendations of <2 g/day [6]. In 2017–2018, the National Health Survey (NHS) found that 1 in 3 people aged 18 years or over had high blood pressure (34%) [7]. In the same years, CVD was the primary cause of death in 41,800 cases (26% of all deaths), and an associated cause of death for 70,600 individuals. This resulted in CVD being the leading cause of death in 2018 (coronary heart disease and cerebrovascular disease combined).

As a result of the extent of the problem, there has been increased focus on dietary salt reduction strategies. This has been driven predominantly by the Global Action Plan on salt, an agreement made among all Members of State of the World Health Organisation to reduce the global sodium consumption by 30% by 2025 [6,8]. The Australian food industry has made several attempts to achieve this, via salt reduction or replacement with other flavour enhancers for a variety of food products.

A 2017 meta-analysis of salt reduction strategies illustrates the different ways in which salt reduction strategies can be implemented in food products. In the analysis, there was no difference in consumer acceptability towards salt-reduced food products such as meats, soups, and miscellaneous products; however, there was a linear trend in consumer acceptability and salt reduction in cheese products [9]. Similar results were illustrated for salt replacement strategies in the same meta-analysis, with studies showing no significant differences in consumer acceptability towards the replacement of NaCl. Potassium chloride (KCl) was commonly used with successful results at varying levels in cheese (up to 40% replacement), meat (10–50% replacement), and ready-made lasagne meal (48% replacement) products [9,10].

Potassium is also a major cation in the human body, as it is the main cation for intracellular fluids and is a large component of lean body mass [1]. The most abundant dietary sources of potassium are in the form of organic acid salts such as potassium phosphate, sulphate, and citrate [11]. Food sources of dietary potassium salts include leafy green vegetables, vine fruits, and root vegetables. In the Australian diet, potatoes (8%), dairy milk (6%), cereal-based mains (6%), coffee or coffee substitutes (5%), and tropical and subtropical fruit juices (5%) are also major sources of dietary potassium [12]. Food sources of potassium are not in the form of potassium chloride, although this is becoming increasingly popular as a substitution for NaCl in food manufacturing via recipe reformulation efforts [13].

Recent research suggests that an increase in the intake of naturally occurring dietary potassium, in the form of increased fruit, vegetable, and dairy product intake, may have a positive impact on reducing blood pressure, decreasing the risk of CVD, and mitigating the consequences of high sodium consumption [12,14]. In the 2014 P.U.R.E. study, there was a strong inverse association between high potassium excretion (a proxy for high dietary intake) and blood pressure, which was even more pronounced in subjects with hypertension [15].

Although an increased dietary potassium intake is beneficial for stroke and cardiovascular health [16], if kidney function is compromised, the accumulation of potassium in the serum can occur (hyperkalemia) [17]. Hyperkalemia can be fatal [18].

Recipe reformulation has been one of the main strategies used to passively reduce sodium consumption by substituting NaCl for KCl in food products so as to reduce the sodium composition of foods. This approach is considered by some to be the logical way forward to reduce the population consumption of sodium [8]. Food products such as breads, processed meats, cheeses, and simmer sauces are some of the food groups in which KCl has been added in an effort to meet the sodium targets for the food industry [19].

Bread is a particularly attractive target for KCl addition, as it is a staple food item in most Western households and is commonly eaten by most age groups, including in Australia [20]. Cereal and cereal-based products are considered one of the main processed food sources in which most people obtain dietary sodium, bread being the highest contributor in this group [9,10]. In 2011–2012, the AHS found that 97% of people reported consuming cereals and cereal-based products, including bread and bread rolls (66%). More recently, in Victoria, bread and bread-based products were found to be the leading dietary sources of sodium overall [10].

While efforts to reduce sodium consumption are driven by the need to reduce the burden of CVD, there are specific subgroups of the population that may be harmed by this approach. This includes people with chronic kidney disease. In Australia, approximately 1 in 10 (1.7 million) Australians aged 18 years and over have CKD [21]. For those individuals with CKD who have progressed to moderate- to severe-CKD, the dietary recommendation is to limit dietary sodium and potassium intake to ~100 mmol/day and ~1 mmol/kg/IBW/day, respectively [22,23].

To date, no dietary modelling studies have evaluated the impact of the addition of KCl in place of NaCl in bread products in the Australian food supply. The aims of this research are three-fold. Firstly, to evaluate the impact of replacing 20%, 30%, and 40% of sodium chloride with KCl in bread and bread-based products consumed by Australians, including Australians with CKD. The second aim is to determine if the replacement of sodium chloride with KCl in bread products will exceed safe limits of consumption for individuals with CKD. Finally, we aim to provide recommendations regarding the safety of replacing sodium with KCl in the Australian food supply.

## 2. Materials and Methods

This study is a diet modelling study using data from the Australian Health Survey from 2011–2013 [24]. To achieve this, we utilised dietary data collected from the 2011–2012 National Nutrition and Physical Activity Survey (NNPAS), a component of the larger Australian Health Survey (AHS) of 2011–2013. A more detailed explanation of the AHS and NNPAS has been previously published [24]. Briefly, the NNPAS is a national survey that gives insight into the dietary and physical activity habits of Australians.

The NNPAS component of the AHS provides dietary information from 9519 randomly selected households, of which information was gathered for one adult and one child (aged 2–17 years) where appropriate. This resulted in 12,153 participants, each of which were asked to provide dietary intake information via two 24-h recalls conducted by trained staff [24]. The participants were asked to provide intake information for all food, beverage, and supplement consumption for the previous 24 h. Interviewers utilised the automated multiple-pass method of questioning for data collection [19]. The research in the present study utilised the available confidential unit record files (CURF) from the 2011–2012 NNPAS. All data from the CURF were managed in Microsoft Excel 2016 in preparation for analysis through the multiple source method (MSM).

In this study, information on food and nutrient intake collected from 24-h recalls were used. Data on food and nutrient intakes were analysed using the 2011–2013 AUSNUT food and nutrient database [24]. The AUSNUT database uses a nested hierarchical structure (major, sub-major, and minor) for each food group, which was utilised for this study [25].

Additional information collected included body weight (kg), height (cm), body mass index (kg/m^2^), systolic blood pressure (mmHg/L), diastolic blood pressure (mmHg/L), and self-reported medical condition status (e.g., cardiovascular disease, diabetes, and chronic kidney disease) were used in the secondary analysis.

To determine the individual amount of bread and bread-based products consumed in the NNPAS 2011–2012, a database was created and then applied to the collected dietary information. For this database, “bread and bread-based products” were identified by utilising the three-digit classification system, in which sub-major food groups ‘”regular breads, and bread rolls (plain/unfilled/untopped varieties)” and “English-style muffins, flat breads, and savoury and sweet breads” were used. In total, 199 of 5740 food items were included. The exclusion of the remaining sub-major groups was not applicable, as they consisted of cereals and cereal-based products, not bread.

The remaining products identified as bread or bread-based products were separated from the remaining food items in the AUSNUT database, and the total sodium and potassium amounts were calculated. This was achieved by dividing the sodium and potassium values in AUSNUT (100 g of food) to show amounts for 1 g of food. These results were then recalculated to accommodate for the direct substitution of potassium for sodium in the identified food items. Potassium chloride was substituted at levels of 20%, 30%, and 40% for sodium chloride. These values were based on previous work, which identified that the palatability of these volumes was acceptable [26].

The bread and bread-based products were then identified in the NNPAS data, and the new total values for the sodium and potassium content were applied at varying levels of substitution. The data were then separated into days 1 and 2 to calculate the total consumption of potassium and sodium from bread or bread-based products on each day. These new totals were used to calculate the daily totals of potassium and sodium for each participant for each day.

The multiple source method (MSM) was used to calculate the usual intakes of dietary potassium based on the data from both days of 24-h recall assessments [27]. The details of this method have been published elsewhere [27,28]. Briefly, MSM is an online statistical tool that employs a three-step procedure using regression models [28,29]. Covariates such as age and gender were accounted for within the regression models, as they are considered to be influential for dietary intake.

To calculate the impact of KCl substitution on people with CKD specifically, we assumed all people who self-reported CKD had moderate CKD or worse (i.e., CKD stage 3b or lower). This was based on evidence that only 10% of the survey respondents with biomedical signs of CKD in the AHS were aware they had the condition, suggesting those who were aware of the condition had more severe forms of the disease [21]. To determine the proportion of adults consuming dietary potassium above safe limits, we undertook three steps. The daily dietary potassium intake was converted to mmol. For those who had a BMI in the healthy weight range, the actual body weight was used to determine intake of potassium in mmol/kg. An adjusted body weight, equivalent to a body weight at a BMI of 27 kg/m^2^ was used for overweight or obese individuals, and then the intake in mmol/kg was determined. For those with no height or BMI in the normal range, the actual weight was used to calculate the mmol/kg. The safe limit was determined to be <1 mmol/kg of K based on evidence-based guidelines [22]. Participants who did not have a listed weight, height, or BMI were excluded from the analysis.

Any implausible values were removed from the data prior to statistical analysis. Data normally distributed were reported using mean and standard deviation (SD), or if not normally distributed, were reported as median and interquartile range. Analysis of the potassium intake via bread and bread-based products was completed using IBM SPSS Version 25.0 [30]. Independent t-tests were used to compare the characteristics of individuals with CKD compared with those without CKD. Chi-square tests were used to examine the differences in proportions between participants.

Ethics approval was not required as the current study was a secondary analysis of the NNPAS 2011–2012 data. Permission was granted to use and analyse the ABS data.

## 3. Results

From the NNPAS 2011–2012, the data from 12,153 participants were utilised, of which 1.3% (*n* = 154) were identified as having CKD. As seen in Table 1, of the 12,153 participants, the majority were females (*n* = 6451; 53.1%). Similarly, the majority of participants who reported that they had CKD were also female (*n* = 87; 56.5%).

The participants with CKD were found to be older than all of the participants (55.9 years (SD 18.9) for CKD; 39.8 years (SD 22.7)) and slightly more overweight than all of the participants. However, both groups were above the healthy weight range for adults (mean weight 75.5 kg (SD 20.9), Body Mass Index (BMI) 27.6 kg/m^2^ (SD 5.9 kg/m^2^) for the CKD group; mean weight 71.5 kg (SD 23.6 kg), BMI 26 kg/m^2^ (SD 6.1 kg/m^2^) for all participants). The mean blood pressure (BP) was also higher in those with CKD than in the larger group, with both systolic BP (128 mmHg/L (SD 25.9); 114 mmHg/L (SD 33.7) for all participants) and diastolic BP (77 mmHg (SD 15.12); 71 mmHg (SD 20.7)) above the healthy range for participants with CKD.

An analysis of the dietary data indicated a large number of participants consumed bread or bread-based products on one or both days (Table 1). Interestingly, a greater percentage of people with CKD reported consuming bread and bread-based products over both days compared with all participants (Table 1; 43.5% vs. 36.5%, *p* = 0.07).

Individuals with CKD consumed more dietary potassium than the larger cohort of all participants (Table 2). Even with no KCl substitution, participants with CKD were consuming 2.76% more potassium than the larger group of all participants (Table 2; 2351.8 ± 888.6) mg/day for CKD; 2288.6 ± 735.2) mg/day for all participants, *p* = 0.37). The contribution of daily potassium from bread and bread products increased from 4.18% (all participants) and 4.29% (CKD) with no substitution to 5.75% (all participants) and 5.91% (CKD) at the level of 40% KCl substitution.

Table 2 also shows the amount of potassium obtained from bread and bread-based products. Again, participants with CKD consumed more dietary potassium from bread products than those without CKD (baseline intake 101 ± 40.1 mg per day compared with 95.6 ± 42.0 mg). Potassium from bread products represents 4.29% of the daily dietary K intake in those with CKD and 4.18% of daily dietary K intake in the larger group of all participants.

At baseline, bread and bread products contributed approximately 14.88% of the daily sodium intake in the larger group, compared with 15.78% of the daily sodium intake in those with CKD (Table 2). The total reduction in the dietary sodium intake achieved from potassium chloride substitution was 5.38% for all participants and 5.68% for those with CKD.

As seen in Table 3, an increasing proportion of individuals with CKD exceeded the safe limits for daily dietary K consumption, although this did not differ significantly (*p* = 0.98). At baseline, 31% exceeded their safe limit of dietary potassium from the overall diet (*n* = 40). This increased to 31.8% of participants at 20% substitution, 32.6% at 30% substitution, and 33% at 40% substitution.

## 4. Discussion

This diet-modelling study explored the impact of the reformulation of bread and bread-based products via salt replacement with KCl. Using the 2011–2012 NNPAS dietary data, we were able to determine the impact and safety of salt replacement on the Australian population, focusing on chronic kidney disease (CKD) patients.

To our knowledge, this is the first diet-modelling study to investigate NaCl replacement with KCl in bread and bread-based products in the Australian food supply, and it is one of the first to investigate salt replacement impacts at a population level.

There were two key findings in this study. Firstly, we evaluated the impact of replacing 20%, 30%, and 40% of NaCl with KCl in bread and bread-based products consumed by Australians, including Australians with CKD. Overall, we found that the dietary potassium intake was higher in those with CKD, particularly from bread and bread products. The impact of substituting varying levels of KCl in bread and bread products resulted in a substantial, but not statistically significant, increase in dietary potassium and a significant ~5% decrease in dietary sodium intake (113.5 mg per day).

Secondly, we found that one in three individuals with CKD exceeded the safe limits of consumption with no substitution, which continued to increase with varying levels of KCl substitution. Sodium reduction and replacement strategies have been encouraged since 2007 on a voluntary basis [8], yet currently, there is no method to track which food manufacturers are involved in this process. This poses a substantial health threat to those susceptible to hyperkalaemia, especially individuals with moderate- to advanced-CKD, as there may be more passive potassium substitution in other processed foods. If this is the case, the potassium intake could increase beyond the safe limits for these individuals, putting them at further risk. Lindberg et al. [19] found that 33% of Australia’s 100 largest food manufacturers make product lines with salt reduction. Of these, more than half (*n* = 17) disclosed that they were actively reducing the NaCl content in at least one of their food products [19]. It is unknown how these reductions occur (whether via substitution or direct reduction), and consumer awareness of this practice is likely to be low.

Despite this, projects to replace discretionary salt with a salt substitute (20–30% KCl) continue to occur, and in some settings have shown widespread success. The outcomes from the Salt Substitute and Stroke Study [31] indicate that by reducing discretionary NaCl with KCl, approximately 450,000 CVD deaths were prevented, and there were approximately 21,000 fewer deaths from CVD annually for individuals with CKD. However, for individuals with CKD, increased potassium intakes were estimated to result in approximately 11,000 additional deaths from CVD, with 42% of these deaths resulting from hyperkalaemia in those with kidney failure [31]. Modelling data from the United Kingdom also suggest that a substitution of KCl in the food supply of up to 25% of sodium may lead to a 2.2-fold increase in hospital presentations for potentially fatal hyperkalemia [32]. There is also concern due to evidence that people without kidney impairment are also at risk of developing dietary induced hyperkalemia [33].

Despite our best efforts, there were some limitations to this study. The 2011–2012 NNPAS was pivotal in understanding the dietary patterns of the Australian population. However, the data were collected in 2011–2012 and may not reflect the current population’s food choices and dietary patterns. We focused solely on bread and bread-based products and did not estimate the contribution of potassium from other food sources. From the NNPAS data, 43.5% of energy was derived from carbohydrates, lower than the recommended 45–65% [34,35]. This is reflective of the popularity of the low carbohydrate diet of the early 2010s. However, recent public health campaigns suggest that the amount of bread and bread-based products consumed will have increased over time [36].

Unfortunately, only 63.6% of participants provided data for two separate days. This left single short-term measures for one-third of the participants, which may have limited the generalisability of the findings. Potential under-reporting of intake is also another important limitation of this study. Analysis of the NNPAS data using the Goldberg cut off method [37] indicated under-reporting was present in the survey for 17% of males and 21% of females [38]. There is also the possibility that under-reporting in the NNPAS does not impact all food products or nutrients equally. Unfortunately, the impact of under-reporting in the present study is unknown. Furthermore, we were unable to ascertain if those with CKD in the NNPAS had received dietary counselling in order to reduce dietary potassium or specific food product intake. Another limitation of this study was that we did not access biomedical information to classify the CKD status of all participants more accurately. This would have enabled a more accurate estimate of the impact of salt replacement in those with various stages of CKD.

We have several recommendations as a result of the study findings. Firstly, widespread sodium replacement with KCl requires the implementation of mandatory food labelling. This is essential for allowing individuals susceptible to hyperkalaemia to make informed food choices. This practice is already in place in the United States, with labelling laws around potassium and calcium changing in 2018 [39]. These labelling changes would also enable health professionals to provide accurate education in order to maximise the dietary potassium intake in people where additional potassium is beneficial.

The second recommendation that arises from this research is to explore the acceptability and palatability of sodium replacement strategies in more detail. For example, consumer acceptability is the primary concern of the food industry [8]. Understanding consumers’ perspectives regarding KCl replacement in staple foods, especially in groups at risk of hyperkalemia, would be valuable and may help inform targets for future salt replacement. It is known that slow, staggered reductions in sodium overtime may be less noticeable and are less likely to impact consumer enjoyment of foods [26]. Meta-analyses of sodium reduction strategies in bread have previously indicated breads with sodium reduction without replacement (in the range of 25–37% reduction in sodium) were just as acceptable to consumers as the original products [9]. Similarly, strategies that reduced sodium in bread by 25% were also acceptable and did not increase the potassium intakes [40]. The third recommendation is for additional dietary modelling using more recent population-level data of dietary patterns.

Population level efforts to reduce the amount of NaCl available in the Australian food supply will help Australia contribute to the WHO global sodium reduction goal of 30% by 2025. However, strategies that include KCl substitution require careful consideration, as many core foods may be impacted. Furthermore, without mandatory food labelling of potassium, this strategy can adversely impact the health of people with CKD, with potentially fatal consequences.

## Figures and Tables

**Table 1 nutrients-13-02472-t001:** Characteristics of the participants.

	All Participants (*n* = 12,153)	Participants with CKD (*n* = 154)	*p* Value
Age mean (SD)	39.8 (22.7)	55.9 (18.9)	<0.001
Gender (% female)	53.1	56.5	0.40
Weight (kg)mean (SD)	71.5 (23.6)	75.5 (20.9)	0.01
Body Mass Index kg/m^2^ mean (SD)	26 (6.1)	27.6 (5.9)	0.06
Systolic blood pressure mm Hg, mean (SD)	114 (33.7)	128 (25.9)	<0.001
Diastolic blood pressure mm Hg, mean (SD)	71 (20.7)	77 (15.1)	<0.001
Number who consumed no bread products on survey days, number (%)	2204 (18.1)	21 (13.6)	0.15
Number who consumed bread products on one survey day, number (%)	5510 (45.3)	66 (42.9)	0.54
Number who consumed bread products on both survey days, number (%)	4439 (36.5)	67 (43.5)	0.07

**Table 2 nutrients-13-02472-t002:** Estimated dietary sodium and potassium intake from bread and bread-based products at varying levels of substitution ^†^.

	All Participants (*n* = 12,153)		Participants with CKD (*n* = 154)	
	Daily Intake Mean (sd)	Daily Intake from Bread Products Mean (sd)	Daily Intake Mean (sd)	Daily Amount of K from Bread Products mg/day, Mean (sd)
Potassium mg/day (no substitution)	2288.6 (735.2)	95.6 (42)	2351.8 (888.6)	101 (40.1)
Potassium mg/day (20% substitution)	2307.8 (738)	114.7 (50.4)	2371.9 (886.9)	121.2 (48.2)
Potassium mg/day (30% substitution)	2317.4 (739.5)	124.3 (54.6)	2382 (886.1)	131.3 (52.2)
Potassium mg/day (40% substitution)	2327 (741.1)	133.8 (58.9)	2392.1 (885.2)	141.4 (56.2)
Sodium mg/day (no substitution)	1906.7 (682.7)	283.8 (116.2)	1807.2 (536.7)	285.1 (100.8)
Sodium mg/day (20% substitution)	1849.7 (671.3)	227 (93)	1750.3 (526.5)	228.1 (80.6)
Sodium mg/day (30% substitution)	1821.2 (666.2)	198.7 (81.3)	1722 (521.8)	199.6 (70.5)
Sodium mg/day (40% substitution)	1792.8 (661.5)	170.3 (69.7)	1693.6 (517.1)	171.1 (60.5)
Percentage of daily K intake (%)	-	4.18	-	4.29
Percentage of daily K intake (%) with 20% substitution of KCl	-	4.97	-	5.11
Percentage of daily K intake (%) with 30% substitution of KCl	-	5.36	-	5.51
Percentage of daily K intake (%) with 40% substitution of KCl	-	5.75	-	5.91
Percentage of daily Na intake (%)	-	14.88	-	15.78
Percentage of daily Na intake (%) with 20% substitution of KCl	-	12.27	-	13.03
Percentage of daily Na intake (%) with 30% substitution of KCl	-	10.9	-	11.59
Percentage of daily Na intake (%) with 40% substitution of KCl	-	9.50	-	10.10

^†^ Data based on MSM-adjusted values from two 24-h recalls from the National Nutrition and Physical Activity Survey (NNPAS) 2011–2012.

**Table 3 nutrients-13-02472-t003:** Analysis of participants with Chronic Kidney Disease (*n* = 129) exceeding the safe limits for dietary potassium at varying levels of potassium chloride (KCl) substitution.

	Baseline (No Substitution)	With 20% Substitution of KCl	With 30% Substitution of KCl	With 40% Substitution of KCl	*p* Value
Number exceeding safe limits	40	41	42	43	0.98
Proportion (%)	31	31.8	32.6	33	

## Data Availability

The data presented in this study are available upon request from the corresponding author.

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
