# Peer review of "Dietary Modelling to Explore the Impact of Potassium Chloride Replacement for Sodium in Bread for Adults with Chronic Kidney Disease"

_nutrients, 2021, doi:10.3390/nu13072472_

Round 1
Reviewer 1 Report
The Australian Health Survey: Nutrition First Results – Foods and Nutrients publication contains food and nutrient information from a 24-hour dietary recall and information on selected dietary behaviours by age groups and sex at the National level.
The authors aim to assess the risk of hyperkalemia when salt in foods is replaced with potassium chloride (KCl) to reduce sodium chloride content.
Attention should be drawn to some possible bias in the study.
Of particular importance in 24-recall nutrition surveys is a widely observed tendency for people to under-report their food intake. This can include:
- actual changes in foods eaten because people know they will be participating in the survey
- misrepresentation (deliberate, unconscious or accidental), e.g. to make their diets appear more ‘healthy’ or be quicker to report.
An NNPAS analysis (https://www.abs.gov.au/statistics/health/health-conditions-and-risks/australian-health-survey-nutrition-first-results-foods-and-nutrients/latest-release#under-reporting) suggests that:
- It is likely that under-reporting is present in both surveys.
- There appears to be an increase in the level under-reporting for males between 1995 and 2011-12, especially for males aged 9 - 50.
- The level of under-reporting by female respondents also appears to have increased, but to a lesser extent than for males.
- In order to achieve an Energy Intake to Basal Metabolic Rate Ratio (EI:BMR) ratio of 1.55 which is the amount required for a normally active but not sedentary population, an increase in mean energy intake of 17% for males and 21% for females is required and greater increases are required for overweight and obese people than those of normal weight.
- Given the association of under-reporting with overweight/obesity and consciousness of socially acceptable/desirable dietary patterns, under-reporting is unlikely to affect all foods and nutrients equally.
In addition to the risk of underreporting, did the authors consider that patients with CKD frequently use special dietary products whose protein and potassium content is reduced? It is possible that this affects the actual potassium intake in these patients.
As the authors correctly pointed out the different stages of renal failure require different dietary prescriptions. Not having the data on what stage of CKD the patients were at makes the data much less plausible.
Author Response
Thank you to the reviewer for the feedback on this manuscript. We agree with the reviewers’ comments about this risk of obtaining biased information from this survey. To address this concern, we have now specifically included reference to under-reporting in the NNPAS and how this adds to the limitations of the study reported (line 292-297).
Re the use of special dietary products: No consideration was made regarding the use of special products. This is because in Australia and New Zealand reduced protein products such as low protein biscuits or bread are not available except to children with inherited metabolic disorders via medical prescription. Reduced potassium products are also not available and K is not labelled on foods. We have included an additional sentence in the limitations that we do not know whether those with CKD receive dietary counselling to reduce dietary potassium intake (line 297-298).
Reviewer 2 Report
Authors reported the impact of KCl substitution in for NaCl in staple food for those with CKD using dietary modelling. Although the research question itself is interesting and methodology seems reasonable, the current manuscript needs full revision according to the comments below.
1) First of all, authors consider KCl is good for health, however, reported merits of K in health stems from K derived from natural food. K from natural food is mostly K as an organic acid salts such as K citrate but not KCl. K salts of organic acid is totally different from KCl. Former is weak alkali, whereas KCl is neutral but relative acid for blood (since blood pH is 7.4 and weak alkali), which is same as NaCl. In addition, most K distributes intracellular space since organic acid salts tend to be transferred intracellular space, whereas KCl tends to be confined to extracellular space since Cl is not transferred intracellularly. Many research shows the critical difference between KCl and K organic acid in terms of the impact on health.
Thus, I doubt the premise that KCl substitution for NaCl is good for health and I would revise the background and would discuss more on this matter.
2) If K substitution for Na does not impact total K load, does that mean K substitution by bread is not enough for Na replacement? Reader would be eager to know how much of NaCl can be reduced by this substitution. Authors should offer this calculation and discuss.
Author Response
1.The reviewer is correct in their description of the differences in forms of potassium in food versus additives. We have not however suggested KCl is good for health. Rather we have indicated that KCl is a popular approach to reduce dietary salt intake and increase dietary potassium at the same time (page 2 line 72: “Potassium chloride is increasingly popular as a substitution for NaCl in food manufacturing via recipe reformulation efforts [12] ). We have included an additional sentence in the discussion quantifying this negative impact: “Modelling data from the United Kingdom suggests that substitution of KCl in the food supply of up to 25% of sodium may lead to a 2.2 fold increase in hospital presentations for potentially fatal hyperkalaemia [31]” (line 276-280).
- The reviewer has made a good point. We did not report the total contribution of bread to sodium intake. We have therefore calculated this and found that in this study the sodium in bread contributed 14.88% of total daily sodium intake (and 15.78% in those with CKD). The change in total dietary sodium intake at varying levels of substitution was a 5.38% reduction in dietary sodium in all participants); and a 5.68% reduction for those with CKD. This is a significant reduction and the impact on dietary sodium intake is now shown in a revised version of Table 2. Additional wording has also been included : “The impact of substituting varying levels of KCl in bread and bread products resulted in a substantial but not statistically significant increase in dietary potassium and a significant ~5% decrease in dietary sodium intake (113.5mg per day)” (line 254-255). We hope this is sufficient for the reviewer.
Round 2
Reviewer 1 Report
The authors responded to my comments
Author Response
No changes are required
Reviewer 2 Report
Authors responded almost appropriately to the comments. However, in the Introduction, there are still sentences which misleads that KCl is good for health, as shown below. This sentences should be deleted or revised not to mix the result of dietary K (K salts) and K additives (KCl) correctly, otherwise this manuscript cannot be accepted.
Recent research suggests the increase in dietary potassium may have a positive impact on reducing blood pressure, decreasing the risk of CVD, and mitigating the consequences of high sodium consumption [11,13]. In the 2014 international P.U.R.E. survey, there was an inverse correlation between increased dietary potassium intake and reduction in blood pressure, even more so in subjects with hypertension [14]. Although increased dietary potassium intake is beneficial for vascular health,
Author Response
We appreciate this point made by the reviewer, and have now made reference to the distinction between naturally occurring forms of potassium and KCl. In addition, to respond directly to the reviewer’s concerns, we have modified the wording in the introduction to read:
Potassium is also a major cation in the human body as it is the main cation for intracellular fluids and is a large component of lean body mass [1]. The most abundant dietary sources of potassium are in the form of organic acid salts such as potassium phosphate, sulphate and citrate [11]. Food sources of dietary potassium salts include leafy green vegetables, vine fruits, and root vegetables. In the Australian diet, potatoes (8%), dairy milk (6%), cereal-based mains (6%), coffee or coffee substitutes (5%), and tropical and subtropical fruit juices (5%) were also major sources of dietary potassium [12]. Food sources of potassium are not in the form of potassium chloride, although this is increasingly popular as a substitution for NaCl in food manufacturing via recipe reformulation efforts [13] .
Recent research suggests an increase in intake of naturally occurring dietary potassium in the form of increased fruit, vegetable and dairy product intake may have a positive impact on reducing blood pressure, decreasing the risk of CVD, and mitigating the consequences of high sodium consumption [12,14]. In the 2014 P.U.R.E. study, there was a strong inverse association between high potassium excretion (a proxy for high dietary intake) and blood pressure, which was even more pronounced in subjects with hypertension[15].
Although increased dietary potassium intake is beneficial for stroke and cardiovascular health [16], if kidney function is compromised, accumulation of potassium in the serum can occur (hyperkalemia) [17]. Hyperkalemia can be fatal [18].
We hope this adjustment to wording is satisfactory for the reviewer and reduces the perception that KCl is good for health.